## Research Article

fear of happiness; intolerance of uncertainty; anxiety; Lebanon

**Corresponding author:**
Sahar Obeid;
Email: saharobeid23@hotmail.com

F.F.-R., S.H. and S.O. are last coauthors.

# Mediating effect of fear of happiness between intolerance of uncertainty and anxiety among Lebanese adults

Laetitia Gerbaka[1], Diana Malaeb[2], Amira Mohammed Ali[3], Fouad Sakr[4], Mariam Dabbous[4], Feten Fekih-Romdhane[5,6] [iD], Souheil Hallit[7,8] [iD] and Sahar Obeid[9] [iD]

[1]Lebanese American University, Lebanon; [2]Gulf Medical University, UAE; [3]Alexandria University, Egypt; [4]Lebanese International University, Lebanon; [5]Université de Tunis El Manar, Tunisia; [6]Razi Hospital, Tunisia; [7]Holy Spirit University of Kaslik Faculty of Medicine and Medical Sciences, Lebanon; [8]Applied Science Private University, Jordan and [9]Lebanese American University - Byblos Campus, Lebanon

## Abstract

Fear of happiness represents the negative feelings that emerge as one apprehends or experiences the positive emotion of happiness. This experience is intrinsically related to intolerance of uncertainty, the apprehension of the unknown, and symptoms of anxiety. While all of these factors are common among the Lebanese population, especially given the hardships it has been through for the past few years, no research has yet studied all three of them in Lebanon. Therefore, this paper tackles the role of fear of happiness as a mediator between intolerance of uncertainty and anxiety among Lebanese adults. The present study is a cross-sectional investigation that recruited 905 Lebanese adults, of which 60% were women, with a mean age of 29.90 years. Fear of happiness partially mediated the association between prospective and inhibitory anxiety and anxiety; higher prospective/inhibitory anxiety was significantly associated with higher fear of happiness and directly associated with higher anxiety. Finally, fear of happiness was significantly and directly associated with higher anxiety. Interventions such as cognitive behavioral therapy and educational programs that tackle these factors may be beneficial to these individuals to relieve symptoms of anxiety and to tackle other negative thought patterns.

## Impact statement

This study provides novel evidence on the psychological mechanisms underlying anxiety among Lebanese adults by examining the mediating role of fear of happiness in the relationship between intolerance of uncertainty and anxiety. By focusing on a construct that is rarely explored in non-Western contexts, the findings extend existing literature beyond Western populations and highlight the cultural relevance of fear of happiness in settings marked by chronic uncertainty and adversity. Giving the ongoing socioeconomic and political challenges in Lebanon, these results have important clinical and public health implications. Addressing fear of happiness and intolerance of uncertainty through targeted psychological interventions and educational programs may help reduce anxiety symptoms and maladaptive cognitive patterns. Overall, this study underscores the importance of culturally informed mental health research and interventions in high-stress contexts.

## Introduction

Anxiety refers to the apprehension of a potential future threat (American Psychiatric Association, 2013; Huang *et al.*, 2021). It is undeniable that anxiety rates among adults across all its levels are higher than ever, sometimes almost reaching 30% of this population (Remes *et al.*, 2016; Mirzaei *et al.*, 2019; Goodwin *et al.*, 2020; Amu *et al.*, 2021; Kan *et al.*, 2021). Symptoms of anxiety include a cognitive aspect, such as the anticipation of the worst or a fear of dying or of losing control; an emotional aspect, such as feeling worried, nervous, terrified or unable to relax; and a physiological aspect such as tingling, dizziness, shakiness, heart palpitations or difficulty breathing (Schwartz *et al.*, 1978; Beck *et al.*, 1988; Koksal and Power, 1990; Spitzer *et al.*, 2006; American Psychological Association, 2011; Ganson *et al.*, 2021; Huang *et al.*, 2021). Anxiety has a considerable negative effect on adult functioning (Adwas *et al.*, 2019), sleep (Cox and Olatunji, 2016; Deng *et al.*, 2021; Chellappa and Aeschbach, 2022; Zhang *et al.*, 2022), attention (Spalding and Nicholls, 2021; Kennedy *et al.*, 2022; Kwarteng *et al.*, 2022) and social skills (Pereira-Lima and Loureiro, 2015;

Moeller and Seehuus, 2019). Because the concerning levels of anxiety among adults have various detrimental consequences on one's life and functionality, it is important to uncover some of its causal factors, in order to explore this psychopathological entity and to tend to its consequences more efficiently (Hashempour and Mehrad, 2014; Maloney *et al.*, 2014; Khan *et al.*, 2017; Millroth and Frey, 2021; Urzúa *et al.*, 2021; Liu *et al.*, 2024). As anxiety translates into severe worry and negative anticipation about the future, intolerance of uncertainty could be a potential leading factor for its occurrence.

### Intolerance of uncertainty and anxiety

Intolerance of uncertainty is defined as the negative "cognitive, behavioral and emotional reactions to uncertainty" (Freeston *et al.*, 1994), considering that uncertainty represents a lack of information about the future (Gu *et al.*, 2020). Individuals who have a high intolerance of uncertainty would highly apprehend potential future negative events and would consider them as a threat, disregarding their likelihood and probability (Carleton *et al.*, 2007; Gu *et al.*, 2020). Intolerance of uncertainty is founded in significant worry about uncertainty in the future and has initially appeared in studies related to anxiety (Carleton, 2012; Carleton *et al.*, 2012; Mahoney and McEvoy, 2012; McEvoy and Mahoney, 2012; Chen *et al.*, 2018; Gu *et al.*, 2020; Jenkinson *et al.*, 2020; Li *et al.*, 2020; Becerra *et al.*, 2023). In fact, intolerance of uncertainty has now become a key factor to distinguish individuals who suffer from generalized anxiety disorder from healthy ones, as well as other anxiety disorders, for example social anxiety (Carleton, 2012; Mahoney and McEvoy, 2012; Counsell *et al.*, 2017; Chen *et al.*, 2018; Gu *et al.*, 2020; Becerra *et al.*, 2023). Therefore, high intolerance of uncertainty is related to higher anxiety (Shihata *et al.*, 2017). An interesting example of this could be the COVID-19 pandemic, which significantly increased the levels of anxiety among the general population (Choi *et al.*, 2020; Hyland *et al.*, 2020; Kan *et al.*, 2021; Lakhan *et al.*, 2020; Peteet, 2020; Daly and Robinson, 2022; Zhu *et al.*, 2023). This could be explained by the uncertainty that this pandemic brought about, for example related to health, work, school and more, which many found intolerable (Del Valle *et al.*, 2020; Liyanage *et al.*, 2021; Taquet *et al.*, 2021; Satici *et al.*, 2022; Andrews *et al.*, 2023; Bavolar *et al.*, 2023; Sergi *et al.*, 2023). This thus resulted in recorded increases in different aspects of anxiety, such as death anxiety and a generally more negative affect (Sergi *et al.*, 2023).

### Intolerance of uncertainty and fear of happiness

Fear of happiness (FoH) is the belief that experiencing joy may lead to negative consequences or suffering (Joshanloo, 2024). While the correlation between intolerance of uncertainty and anxiety has been established, FoH seems to mediate the relationship between these two variables. This will be justified by the correlation between FoH and each of these factors, starting with intolerance of uncertainty.

First, it is important to note that little research has been done about intolerance of uncertainty and FoH, highlighting once again the importance of this paper. However, intolerance of uncertainty has been associated with trouble experiencing any kind of positive emotion, thus providing a link to FoH – the difficulty experiencing this positive emotion (Morriss *et al.*, 2023; Sahib *et al.*, 2024; Sriyanto and Hakim, 2024). A study conducted by Bakioğlu *et al.* (2021) has also established a link between intolerance of uncertainty and higher anxiety, and lower positivity, which can be related to levels of FoH.

### Fear of happiness and anxiety

Second, research has noted that FoH can be linked to anxiety among adults (Joshanloo, 2013; Türkmen and Sezer, 2023; Joshanloo, 2024). FoH is increasingly experienced by adults, especially those who suffer from low self-esteem or who have traits such as perfectionism or pessimism, also highlighting the role of culture and religion in this trait (Joshanloo, 2024). Moreover, this factor has been shown to often result not only in depression, as previously noted (Gilbert *et al.*, 2014; Jordan *et al.*, 2021; De Vuyst *et al.*, 2023; Eliüşük Bülbül and Özbay, 2024; Elmas and Çevik, 2024; Joshanloo, 2024), but also in stress, levels of antisocial behavior, panic disorder, cognitive distortions and notably anxiety (Joshanloo, 2013; Gilbert *et al.*, 2014; Muhtar, 2016; Belen *et al.*, 2020; Blasco-Belled *et al.*, 2021; Lambert *et al.*, 2022; Elmas, 2022; İşgör *et al.*, 2022; Türkmen and Sezer, 2023; Elmas and Çevik, 2024; Gandhi, 2024; Joshanloo, 2024; Srivastava, 2024). Indeed, FoH impairs motivation to achieve goals, as well as the self-confidence needed to believe that one is initially capable of achieving these goals, partly explaining the correlation between FoH and increased anxiety levels (Belen *et al.*, 2020). Additionally, a study conducted by İşgör *et al.* (2022) underlined the link between FoH and rumination patterns – frequent and prolonged negative thoughts about one's subjective experiences, personality and emotions (Stelmach-Lask *et al.*, 2024). These are a main aspect of anxiety, aligning with previously mentioned research (Gilbert *et al.*, 2014; Jordan *et al.*, 2021; De Vuyst *et al.*, 2023; Eliüşük Bülbül and Özbay, 2024; Gandhi, 2024) and thus providing another connection between FoH and anxiety. Therefore, the literature indicates that there is a correlation between FoH and anxiety, proven by an increase in symptoms of anxiety and patterns related to it such as rumination, or lower motivation and self-esteem noted among individuals with FoH. Considering that a link has also been established in the literature between FoH and intolerance of uncertainty, while also taking into account the major role of FoH in anxiety, this study considers FoH as a mediator between these two variables.

### Rationale and aim

With the aforementioned high prevalence of anxiety among adults (Remes *et al.*, 2016; Mirzaei *et al.*, 2019; Goodwin *et al.*, 2020; Amu *et al.*, 2021; Kan *et al.*, 2021), the importance of widening the research and literature about this trait is evident. Furthermore, with intolerance of uncertainty and FoH becoming increasingly important and interesting topics in the field of psychology (Carleton *et al.*, 2007; Gu *et al.*, 2020; Morriss *et al.*, 2023; Joshanloo, 2024; Sahib *et al.*, 2024; Sriyanto and Hakim, 2024), the relevance of this paper is highlighted. This importance is also exacerbated by the potential application of these findings in clinical practice and interventions, especially in cognitive behavioral therapy (CBT), be it to tackle the thought patterns behind intolerance of uncertainty and the maladaptive schemas of FoH symptoms of anxiety (Bomyea *et al.*, 2015; Talkovsky and Norton, 2016; Torbit and Laposa, 2016; Keefer *et al.*, 2017; Blasco-Belled *et al.*, 2021; Laposa *et al.*, 2022; Wilson *et al.*, 2023; Elmas and Çevik, 2024; Joshanloo, 2024; Srivastava, 2024; Engin *et al.*, 2025; Yıldırım *et al.*, 2025).

It is also important to note that the link between the three variables intolerance of uncertainty, FoH and anxiety has not yet been clearly researched, especially not considering FoH as a mediation factor, with no studies studying these exact variables together. Indeed, findings have established a link between intolerance of uncertainty and anxiety, as previously mentioned (Carleton,

2012; Mahoney and McEvoy, 2012; McEvoy and Mahoney, 2012; Bomyea *et al.*, 2015; Chen *et al.*, 2018; Gu *et al.*, 2020; Jenkinson *et al.*, 2020; Li *et al.*, 2020; Becerra *et al.*, 2023; Kock *et al.*, 2023; Joshanloo, 2024; Morriss, 2025), as well as between FoH and anxiety (Joshanloo, 2013; Bomyea *et al.*, 2015; Kock *et al.*, 2023; Türkmen and Sezer, 2023; Joshanloo, 2024). However, intolerance of uncertainty includes a higher sensitivity to threat (Morriss *et al.*, 2016; Tanovic *et al.*, 2018; Milne *et al.*, 2019), which can reinforce the FoH schema that perceives happiness and its consequences as a threat (De Vuyst *et al.*, 2023; Elmas and Çevik, 2024; Joshanloo, 2024). This higher sensitivity to threat is also correlated with increased levels of anxiety, which is often represented as irrational worry about potential threats (O'donovan *et al.*, 2013; Nelson *et al.*, 2015; Bardeen and Daniel, 2018; Burani and Nelson, 2020). Moreover, a higher sensitivity to threat or vigilance can also be linked to avoidance or repression of emotions (Bardeen *et al.*, 2015; Li *et al.*, 2020; Joshanloo, 2024), which FoH – the avoidance of the emotion of happiness – is an example of (Elmas and Çevik, 2024; Joshanloo, 2024; Srivastava, 2024; Lee *et al.*, 2025). Higher vigilance and avoidance/repression of emotions are also linked to rumination (Garrison *et al.*, 2014; Brookes *et al.*, 2017; Eisma *et al.*, 2020; Stelmach-Lask *et al.*, 2024). Additionally, the latter is a pattern observed in FoH, as previously shown (İşgör *et al.*, 2022; Stelmach-Lask *et al.*, 2024) as well as a symptom of anxiety. Therefore, this suggests that, through the examples of higher sensitivity to threat, avoidance of emotions and rumination, FoH could be considered as a link between intolerance of uncertainty and anxiety. Finally, a study conducted by Körün and Satıcı (2025) examines the mediating role of FoH between intolerance of uncertainty and rumination. However, rumination has been established as a symptom of anxiety, which thus underlines the aim of this study of broadening these results to anxiety in general, by studying the mediating role of FoH between intolerance of uncertainty and anxiety.

Given that FoH may vary across individuals depending on sociodemographic characteristics, such as sex or alcohol use, these variables will be accounted for in the present study. Indeed, research suggests that high levels of FoH are more often recorded among women than among men, which can be related to greater emotional sensitivity and socialization around emotional restraint (Joshanloo, 2013). In some cultures, women may internalize beliefs that expressing happiness invites negative consequences such as envy or emotional vulnerability (Carbone *et al.*, 2024). These gender differences appear to be shaped by cultural context and individual coping styles, though findings are not entirely consistent across studies. Furthermore, research suggests that FoH may lead some individuals to use alcohol to cope with uncomfortable emotions or avoid positive feelings (Sayette, 2017). However, this relationship is not yet well understood and requires further study.

Furthermore, the significance of studying these variables among Lebanese adults lies in the relevance that all of intolerance of uncertainty, FoH and anxiety have in this population. Indeed, in the past few years, Lebanon has survived an enormous number of crises, from the COVID-19 pandemic to the financial collapse, to the 2020 Beirut Blast, to the recent war (El Othman *et al.*, 2021; Hashim *et al.*, 2022; World Bank, 2024; Altaweel, 2025). This has undeniably impacted their mental health, even going into what Farran (2021) labels as "a mental health epidemic", one of the main mental disorders being anxiety (El-Khoury *et al.*, 2020; Haddad *et al.*, 2020; Hallit *et al.*, 2020; Kmeid *et al.*, 2020; Merhy *et al.*, 2021). Moreover, it is important to note that FoH is culturally wired in the Middle Eastern and Lebanese mentality and values, and this has increased after the recent events (Halawi and Salloukh, 2020).

However, the matter is still widely understudied and is currently arising in the research field (El Khoury *et al.*, 2024). A sample of Lebanese adults was chosen due to the relevance of the research topic within the Lebanese cultural and socioeconomic context. Lebanon's unique social dynamics, exposure to ongoing stressors and evolving mental health awareness make it a valuable population for examining the link between FoH, intolerance of uncertainty and anxiety. Moreover, a scarcity of empirical data from this population in the existing literature was noted, highlighting the need for localized research to inform culturally appropriate interventions and theoretical models. For these reasons, this study aimed at assessing the mediating role of FoH between intolerance of uncertainty and levels of anxiety among a sample of Lebanese adults.

## Methods

### Study design and participants

This study employed a correlational design based on cross-sectional data collection. It collected data from 905 participants in Lebanon between August and September 2024, with eligibility requirements including being a resident or citizen of Lebanon, with a minimum age of 18 years. The mean age of participants was 29.90 years, with a standard deviation of 9.28, and 60% of them identified as female.

Data collection was done using a survey created on Google Forms, which was shared on social media. The research team employed snowball sampling to recruit participants. This technique began with sending the survey link to a few individuals, who were encouraged to share it within their network, facilitating a broader reach. Given Lebanon's context of instability and ongoing crises, traditional recruitment methods may not have been viable, so snowball sampling was used to build trust and overcome potential reluctance to participate.

The survey was anonymous, with participants providing digital informed consent to be able to access the questionnaire. Participation was voluntary and without compensation, and the completion of the survey needed about 20 minutes.

The same methods were used as other studies that also tackled intolerance of uncertainty in Lebanon (Assaf *et al.*, 2025; Obeid *et al.*, 2025) to ensure the validity of the results and the conformity of the procedure.

### Minimal sample size calculation

To detect the mediated effect, a sample size of 411 individuals is needed, calculated using the formula suggested by Fritz and MacKinnon (2007). The formula is given as $n = \frac{L}{f^2} + k + 1$, where $n$ is the minimal sample size. In this context, $L$ is set to 7.85, which applies for a one-predictor ordinary least squares regression with a type I error rate ($\alpha$) of 0.05 and a power of 0.80. The value of $f$, representing a small effect size, is 0.14. The value of $k$ is 9, which denotes the number of predictors in the regression equation.

### Questionnaire

The questionnaire was presented in Arabic, Lebanon's official language, to ensure accessibility for participants. It started with an introductory section outlining the study's purpose and included an online consent checkpoint where participants confirmed their voluntary participation. The consent process assured participants of the confidentiality and anonymity of their responses.

The survey collected various sociodemographic data, such as age, sex, marital status and Household Crowding Index (HCI), which serves as a measure of socioeconomic status (SES). The HCI was calculated by dividing the number of people in the household by the number of rooms in the house (excluding the kitchen and bathrooms). These variables are inversely proportional; thus, a lower HCI suggests a higher SES, as it means less overcrowding at home (Melki *et al.*, 2004). Participants' smoking habits, alcohol use, cannabis consumption and other psychological conditions were also noted, as well as four other self-report scales explained below.

The *Intolerance of Uncertainty Scale (IUS-12)* (Carleton *et al.*, 2007). The shortened version of the IUS-27, validated in Arabic (Chaaya *et al.*, 2024), includes 12 items scored on a Likert scale from 1 (not at all characteristic of me) to 5 (very characteristic of me). This scale measures the extent to which the participant is distressed by uncertainty. It has two subscales: prospective anxiety evaluates the need for predictability and the tendency to actively seek it to reduce the risk of uncertainty, and inhibitory anxiety scores the participants' avoidance of uncertainty and the behavioral inhibition when faced with it. Prospective anxiety is scored by adding the answers of the subscales' items; the higher the score, the greater the intolerance of uncertainty. In this study, Cronbach's α was 0.90 for both subscales.

*Generalized Anxiety Scale-5 (GAD-5)*. Validated in Arabic (Sawma *et al.*, 2024), the GAD-5 is a self-report instrument that consists of five items, each rated on a four-point Likert scale from 0 (not at all) to 3 (nearly every day) (Goldberg *et al.*, 1988). The total score is computed by adding the answers of the items, with higher scores indicating higher anxiety (Cronbach's α in this study = 0.90).

*The Fear of Happiness Scale (FHS)* (Joshanloo, 2013). Validated in Arabic (El Khoury *et al.*, 2024), the FHS is a short questionnaire that includes five items, which are intended to score for FoH, i.e., the belief that the participant does not deserve happiness or that the latter will bring about negative consequences. The items are scored on a Likert scale from 1 to 7, from strongly disagree to strongly agree. The total score is computed by adding the answers of the statements, with higher scores indicating higher FoH.

### Statistical analysis

The SPSS v.27 software was used for the statistical analysis. The anxiety score was considered normally distributed since the skewness and kurtosis values were between the −1 and +1 interval. The Student t-test was used to compare the mean value of a continuous variable across two levels of a dichotomous variable and the Pearson's test to correlate two continuous variables. In the context of this study, effect size refers to a quantitative measure of the strength or magnitude of a relationship or difference between variables, independent of the sample size. A value of 0.2 reflects a weak effect, whereas values of 0.5 and 0.8 indicate a moderate and large effect size, respectively (Sullivan and Feinn, 2012).

The mediation analysis was performed using PROCESS MACRO (a SPSS add-on) v3.4 Model 4, with the number of bootstrap samples set at 5000 and a 95% confidence interval. Four pathways resulted from this analysis: pathway A of the independent variable to the mediator, pathway B of the mediator to the dependent variable and pathways C and C' indicating the total and direct effects of the independent variable to the dependent variable. We considered the mediation analysis to be significant if the confidence interval did not pass by zero. Covariates entered in the model were those that showed a $p < 0.25$ in the bivariate analysis. $P < 0.05$ was considered statistically significant.

### Patient and public involvement

As this study was conducted among the non-clinical general population, there was no direct involvement of patients, service users or caregivers in the design, conduct, analysis or interpretation of the data. However, the study aimed to gather insights from a broad cross-section of the public to better understand general trends and behaviors. While the research did not include active participation from specific patient or public groups, the findings may indirectly benefit from the diverse perspectives of the general population included in the study.

### Results

#### Participants

In total, 905 participants participated in this study, with a mean age of 27.38 years and 60% being female. Other descriptive statistics of the sample are presented in Table 1.

#### Bivariate analysis of factors associated with anxiety

Female vs male, drinking alcohol and having psychological problems were significantly associated with higher anxiety (Table 2). Moreover, higher prospective anxiety, inhibitory anxiety and FoH were significantly associated with higher anxiety (Table 3).

#### Analysis of mediation

The mediation analysis taking anxiety as the dependent variable was adjusted over the following covariates: sex, smoking, alcohol and psychological problems. FoH partially mediated the association between prospective (indirect effect: Beta = 0.17; Boot SE = 0.03; Boot CI 0.10; 0.24) and inhibitory (indirect effect: Beta = 0.23; Boot SE = 0.05; Boot CI 0.14; 0.32) anxiety and anxiety; higher prospective/inhibitory anxiety was significantly associated with higher FoH and directly associated with higher anxiety. Finally, FoH was significantly and directly associated with higher anxiety (Figures 1 and 2).

### Discussion

#### Intolerance of uncertainty and fear of happiness

To start with, both prospective and inhibitory anxiety, which are facets of intolerance of uncertainty, are significantly correlated with FoH. This could be explained by the fact that for some, especially in the Lebanese population, living and accepting happiness may be unfamiliar, as individuals' life satisfaction would be low (Joshanloo, 2013; Abdel-Khalek, 2015; Yildirim, 2019). Indeed, the levels of unhappiness and depression among the Lebanese are high (Ahmadieh *et al.*, 2018; Harper Shehadeh *et al.*, 2020; Obeid *et al.*, 2020; Cuijpers *et al.*, 2022). Therefore, the prospective experience of happiness may thus be worrisome for some, as they do not really know what to expect, increasing their uncertainty about it, leading to a fear of this emotion. Thus, if some people struggle with anticipating new experiences and do not tolerate this much, they may fear happiness, as this emotion may be unfamiliar to them.

**Table 1.** Sociodemographic and other characteristics of the sample (*N* = 905)

| Variable | *N* (%) |
|---|---|
| Sex | |
| Male | 362 (40.0%) |
| Female | 543 (60.0%) |
| Marital status | |
| Single, divorced, widowed | 804 (88.8%) |
| Married | 101 (11.2%) |
| Smoking | |
| No | 506 (55.9%) |
| Yes | 399 (44.1%) |
| Alcohol | |
| No | 784 (86.6%) |
| Yes | 121 (13.4%) |
| Cannabis | |
| No | 851 (94.0%) |
| Yes | 54 (6.0%) |
| Psychological problems | |
| No | 797 (88.1%) |
| Yes | 108 (11.9%) |
| | Mean ± SD |
| Age (years) | 27.38 ± 9.28 |
| Household overcrowding index (person/room) | 1.01 ± 0.43 |
| Anxiety | 17.57 ± 11.38 |
| Fear of happiness | 12.99 ± 6.21 |
| Prospective anxiety | 19.26 ± 5.92 |
| Inhibitory anxiety | 12.84 ± 4.51 |

Moreover, this correlation between intolerance of uncertainty and FoH can also be demonstrated by the hopelessness that some individuals may experience, which is also common in Lebanon given the many hardships this population has been through (Obeid *et al.*, 2019; Chahine *et al.*, 2020; Sharanek, 2020; Obeid *et al.*, 2021; Zakhour *et al.*, 2021). In fact, as one experiences apprehension of the future and its uncertainty, hopelessness about it and the inability to cope with it may arise (Demirtas and Yildiz, 2019; Özdemir *et al.*, 2021; Engin *et al.*, 2024; Erkan and Kavak Budak, 2024). Consequently, this hopelessness may also reinforce the person's FoH as they do not believe that they deserve to or even can experience this emotion in the future anymore, thus losing hope about it and even becoming scared and worried about it (Bloore *et al.*, 2020; Koca, 2020; Dobos *et al.*, 2024). Therefore, intolerance of uncertainty reduced feelings of hope among individuals, who thus have a more negative perspective on their future, leading to an increased FoH.

### *Fear of happiness and anxiety*

Additionally, FoH is also linked to higher levels of anxiety. To start with, an explanation for this could be the reduced self-efficacy that one may experience when they fear happiness and believe that they do not deserve it (Erozkan *et al.*, 2016; Marzi and Saadati Shamir, 2019; Mete, 2021; Türkmen and Sezer, 2023; Gandhi, 2024). In terms, this could increase their anxiety (Bandura, 1991; Razavi *et al.*, 2017; Ng and Lovibond, 2020; Burns *et al.*, 2021; Simonetti *et al.*, 2021; Lau *et al.*, 2022; Wan *et al.*, 2024) as they are not achieving their potential and fulfilling their purpose. Examples of this can be at work (Lestari *et al.*, 2024), in their relationships (Weiser and Weigel, 2016), hobbies (Bum *et al.*, 2021; Neroni *et al.*, 2022), etc., which all stem from the deep fear of being happy and thus may induce anxiety (Regan *et al.*, 2014; Li *et al.*, 2019; Deady *et al.*, 2022; Li *et al.*, 2023). Therefore, FoH creates a circle of self-sabotage in the individual, whose self-efficacy decreases on many levels, in turn increasing their anxiety. This also aligns with the previously stated fact that FoH transforms one's perspective and beliefs about themselves (Stelmach-Lask *et al.*, 2024), morphing them into negative beliefs about oneself, leading them to this cycle of self-sabotage that induces anxiety.

Furthermore, the effect of FoH on anxiety may also be explained by the maladaptive belief patterns that one develops about others and the world, which were previously explained and which may have several significant effects on anxiety (Blasco-Belled *et al.*, 2021; Elmas and Çevik, 2024; Joshanloo, 2024). Indeed, on the one hand, these negative beliefs about others may bring about insecurities in the individual, who may for example start feeling judged by others, unwanted or even unloved, making them withdraw from their social life (Savitsky *et al.*, 2001; Ciarrochi, 2004; Leary *et al.*, 2007; Frías *et al.*, 2014; Skeen, 2014; Valshtein *et al.*, 2020; van Prooijen, 2023; Jørgensen and Bøye, 2024), or feeling paranoid because they think others want to harm them, thus becoming excessively weary about other people. On the other hand, negative beliefs about the world could also bring them to withdraw to protect themselves from all the bad that they believe exists outside, as one may start believing that the world is cruel, unfair and filled with sadness (Cann *et al.*, 2010; Clifton *et al.*, 2019; Wood *et al.*, 2022). For example, an individual who fears happiness and has developed the maladaptive belief that the world is not worth trying will isolate themselves, start spending too much time at home or may quit college, which may create financial, parental or social problems that can increase their anxiety (Clifton and Kerry, 2023). Consequently, the negative beliefs about the world and others that stem from FoH increase anxiety, as the individual becomes dysregulated and their lifestyle is negatively affected in several aspects (Buschmann *et al.*, 2018; Sherwood *et al.*, 2020; Hickman *et al.*, 2021; Leibovitz *et al.*, 2021; Vasileva *et al.*, 2021; Vazquez *et al.*, 2021; Galway and Field, 2023). In addition to that, in their most severe forms, these negative and maladaptive thoughts and emotions might amplify and become more severe, sometimes to the point of developing into severe mental disorders, such as depression, generalized anxiety disorder or social anxiety disorder, eating disorders, borderline personality disorder, or even obsessive-compulsive disorder or posttraumatic stress disorder (Borton *et al.*, 2005; Vocks *et al.*, 2008; Taylor and Stopa, 2013; Koerner *et al.*, 2015; Eagleson *et al.*, 2016; Buschmann *et al.*, 2018; Masland *et al.*, 2020; Hodny *et al.*, 2021; Siregar *et al.*, 2021; Kube, 2023; Palmieri *et al.*, 2023; Kube and Rauch, 2025). Therefore, FoH may also create negative beliefs about the world and others, which may lead to anxiety through social withdrawal, paranoia, deterioration of lifestyle and more, thus producing higher anxiety in the concerned individuals.

**Table 2.** Bivariate analysis of factors associated with anxiety

| Variable | Mean ± SD | t | df | P | Effect size |
|---|---|---|---|---|---|
| Sex | | −4.71 | 849.15 | **<0.001** | 0.310 |
| Male | 15.48 ± 10.19 | | | | |
| Female | 18.97 ± 11.91 | | | | |
| Marital status | | −0.91 | 122.34 | 0.367 | 0.102 |
| Single, divorced, widowed | 17.45 ± 11.27 | | | | |
| Married | 18.60 ± 12.22 | | | | |
| Smoking | | 1.16 | 903 | 0.248 | 0.077 |
| No | 17.96 ± 11.28 | | | | |
| Yes | 17.08 ± 11.51 | | | | |
| Alcohol | | −1.55 | 146.33 | **0.123** | 0.174 |
| No | 17.31 ± 11.03 | | | | |
| Yes | 19.29 ± 13.36 | | | | |
| Cannabis | | −0.02 | 903 | 0.981 | 0.003 |
| No | 17.57 ± 11.44 | | | | |
| Yes | 17.61 ± 10.54 | | | | |
| Psychological problems | | −2.74 | 129.14 | **0.007** | 0.316 |
| No | 17.15 ± 11.09 | | | | |
| Yes | 20.72 ± 12.95 | | | | |

*Note:* Numbers in bold indicate significant *p* values.

**Table 3.** Pearson correlation matrix

| | 1 | 2 | 3 | 4 | 5 |
|---|---|---|---|---|---|
| 1. Anxiety | | | | | |
| 2. Prospective anxiety | 0.47*** | | | | |
| 3. Inhibitory anxiety | 0.43*** | 0.77*** | | | |
| 4. Fear of happiness | 0.38*** | 0.39*** | 0.38*** | | |
| 5. Age | 0.03 | 0.02 | 0.01 | 0.03 | |
| 6. Household overcrowding index | −0.01 | −0.05 | −0.04 | −0.03 | 0.17*** |

*Note:* ***$p < 0.001$.

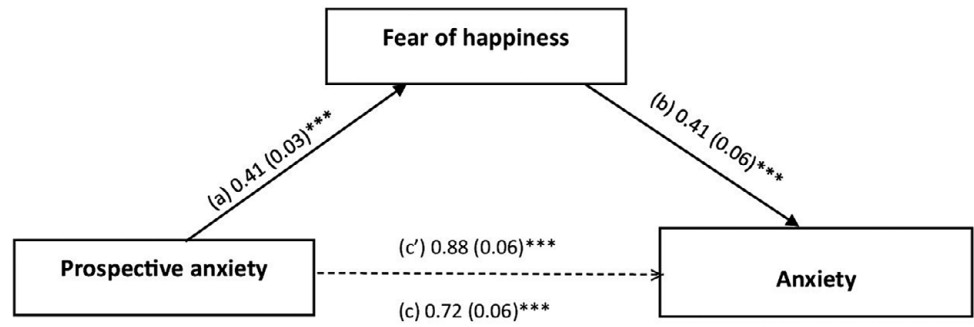

**Figure 1.** (a) Relationship between prospective anxiety and fear of happiness ($R^2 = 0.184$); (b) relationship between fear of happiness and anxiety ($R^2 = 0.284$); (c) total effect of prospective anxiety on anxiety; (c') direct effect of prospective anxiety on anxiety ($R^2 = 0.244$). The numbers represent regression coefficients and their standard errors. ***$p < 0.001$.

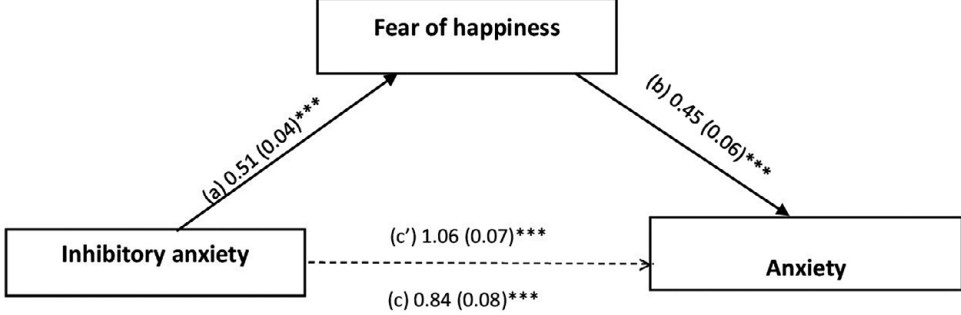

**Figure 2.** (a) Relationship between inhibitory anxiety and fear of happiness ($R^2 = 0.172$); (b) relationship between fear of happiness and anxiety ($R^2 = 0.263$); (c) total effect of inhibitory anxiety on anxiety; (c') direct effect of inhibitory anxiety on anxiety ($R^2 = 0.213$). The numbers represent regression coefficients and their standard errors. ***$p < 0.001$.

## Mediation analysis

In sum, FoH represents a mediator between intolerance of uncertainty and anxiety, as its link with the two variables has been shown in the previous sections, showing that FoH significantly increases the levels of anxiety, also considering intricate relationship with levels of intolerance of uncertainty. Indeed, the latter may bring about FoH since the latter may be an unfamiliar emotion to some (Joshanloo, 2013; Abdel-Khalek, 2015; Yildirim, 2019). FoH may also emphasize a feeling of hopelessness among some subjects (Demirtas and Yildiz, 2019; Bloore et al., 2020; Koca, 2020; Özdemir et al., 2021; Dobos et al., 2024; Engin et al., 2024; Erkan and Kavak Budak, 2024). Consequently, FoH increases levels of anxiety since it may reduce one's feeling of self-efficacy (Regan et al., 2014; Bandura, 1991; Erozkan et al., 2016; Razavi et al., 2017; Li et al., 2019; Marzi and Saadati Shamir, 2019; Ng and Lovibond, 2020; Burns et al., 2021; Mete, 2021; Simonetti et al., 2021; Deady et al., 2022; Lau et al., 2022; Li et al., 2023; Türkmen and Sezer, 2023; Gandhi, 2024; Wan et al., 2024). This may stem from their negative beliefs about themselves, which may thus deteriorate their lifestyle and increase their levels of anxiety. FoH may also develop negative and maladaptive beliefs of others and the world (Blasco-Belled et al., 2021; Elmas and Çevik, 2024; Joshanloo, 2024) and may be very dangerous for their lifestyle (Buschmann et al., 2018; Sherwood et al., 2020; Hickman et al., 2021; Leibovitz et al., 2021; Vasileva et al., 2021; Vazquez et al., 2021; Clifton and Kerry, 2023; Galway and Field, 2023), their social life (Savitsky et al., 2001; Ciarrochi, 2004; Leary et al., 2007; Cann et al., 2010; Frías et al., 2014; Skeen, 2014; Clifton et al., 2019; Valshtein et al., 2020; Wood et al., 2022; van Prooijen, 2023; Jørgensen and Bøye, 2024) and their mental health (Borton et al., 2005; Vocks et al., 2008; Taylor and Stopa, 2013; Koerner et al., 2015; Eagleson et al., 2016; Buschmann et al., 2018; Masland et al., 2020; Hodny et al., 2021; Siregar et al., 2021; Kube, 2023; Palmieri et al., 2023; Kube and Rauch, 2025), which also includes elevated levels of anxiety (Buschmann et al., 2018; Sherwood et al., 2020; Hickman et al., 2021; Leibovitz et al., 2021; Vasileva et al., 2021; Vazquez et al., 2021; Clifton and Kerry, 2023; Galway and Field, 2023).

Therefore, FoH is considered a mediator because of the one-way relationship proposed with anxiety, as no effect of anxiety on FoH has been recorded (Belen et al., 2020; İşgör et al., 2022; Türkmen and Sezer, 2023; Elmas and Çevik, 2024; Joshanloo, 2024; Srivastava, 2024; Teke, 2025). Furthermore, while the cognitive triad – negative and maladaptive beliefs about oneself, others and the world – has been proven to have an influence on depression and depressive symptoms, the main link that has been made to anxiety is that the latter acts as a result of these thoughts and beliefs, often

because of depression (Kaslow et al., 1992; Braet et al., 2015; Berghuis et al., 2020). In fact, this resulting anxiety often presents as rumination, which has been established as a result of maladaptive schemas and negative beliefs about oneself, others and the world (Orue et al., 2014; Balsamo et al., 2015; Carlucci et al., 2018; Bazargani et al., 2023; Eshghifar et al., 2025). This also aligns with the studies that show that maladaptive belief patterns may lead to more severe cases of generalized and social anxiety disorders (Taylor and Stopa, 2013; Koerner et al., 2015; Buschmann et al., 2018). Therefore, a one-way causality between FoH and anxiety can be implied, proving the role of the latter as a mediator between intolerance of uncertainty and anxiety.

## Clinical implications

As a result of these inferences, it is clear that intolerance of uncertainty, through its mediator, FoH, contributes to elevated levels of anxiety among Lebanese adults. Therefore, these negative anxiety symptoms could be dealt with and relieved through the implementation of CBT. It would be advantageous for individuals to cope with this high anxiety in an efficient, time-saving, evidence-based way (DiMauro et al., 2013; Loerinc et al., 2015; Kishita and Laidlaw, 2017; McMorris et al., 2024). Indeed, patients would learn to target their negative beliefs that induce their inability to cope with this anxiety, as well as to practice exercises that reduce the symptoms that they would be suffering from (Flynn and Warren, 2014; Robichaud and Dugas, 2015). Negative and maladaptive beliefs related to intolerance of uncertainty and FoH can also be tackled through CBT (Talkovsky and Norton, 2016; Keefer et al., 2017; Blasco-Belled et al., 2021; Wilson et al., 2023; Elmas and Çevik, 2024; Srivastava, 2024). Additionally, an anxiety prevention program that has been created based on CBT named "Journey of the Brave" could also be implemented, allowing the participants to target their anxiety symptoms and their aforementioned factors (Ohashi et al., 2024). This program is comprised of group therapy sessions that allow improvement of social skills, interpersonal connection and discovery and more, making the participants more open and willing to adapt and implement cognitive and behavioral changes (Ohashi et al., 2024). Therefore, these allow the person to tackle different aspects of their mental health, one of them being their anxiety, which is why this program has been reported to reduce anxiety symptoms (Ohashi et al., 2024). Finally, patient education could be a major intervention that would help in the alleviation or at least understanding of the concerned population in their thought patterns, emotions and symptoms, thus allowing them to improve based on this understanding (Huang et al., 2021).

## Limitations

It is important to start by mentioning that the literature available about the subject was little, especially regarding FoH, which might have jeopardized the discussion about mediation. Indeed, no article targeted this subject directly – considering the three variables together, whether in Lebanon or not – which resulted in little literature to rely on to discuss the mediation and compare the present results to. Therefore, this surely was a significant limitation to the findings of the paper. Furthermore, the data were collected through self-reported questionnaires, making the results at risk of a social desirability bias, if participants gave socially acceptable responses, for example considering their potential FoH. Participants might also have lacked self-awareness, thus reducing the accuracy of results such as anxiety. Thus, this might have affected the accuracy of the data collected and thus of the results, for example about the levels of anxiety. Moreover, the data were collected, and the survey was sent using the snowballing sampling technique, thus reducing the inclusivity of the sample and the reach of different populations and groups. Indeed, since random sampling was not applied due to circumstantial constraints, the generalizability of the results to the general Lebanese adult population may be questionable. Thus, the sample may not represent this population as a whole and may be biased depending on the participants it reached and who completely filled the survey. Additionally, the interpretation of the association between intolerance of uncertainty and FoH should be approached cautiously. In the original discussion, this link was partly interpreted in light of the mean level of FoH in the sample; however, the average score of a variable cannot be used as evidence for causal or explanatory mechanisms underlying its association with another variable. Therefore, the relationship between these constructs should be considered explanatory, with further research needed to clarify the directionality and underlying processes. It is also important to note that a longitudinal study might have been more valid and reliable to test the mediation hypothesis, which would have provided a higher accuracy of results over time, as well as of the causal effect.

## Conclusion

This study showed that FoH mediates the relationship between the two other variables, as FoH may be correlated with intolerance of uncertainty – represented by prospective and inhibitory anxiety – in the sense that happiness may represent a new and unfamiliar emotion to the Lebanese and through the hopelessness that this population may suffer from. Additionally, FoH acts as a mediator between these two variables by being significantly correlated with levels of anxiety, reducing the self-efficacy, productivity and achievement of the individual, while concurrently reinforcing their negative and maladaptive beliefs about themselves, others and the world. Therefore, CBT may represent a beneficial asset and intervention for these individuals. Programs to implement this approach or to educate the population may also be put into effect, benefiting different groups and populations of Lebanese. Ultimately, it would be interesting to mention the effect of low self-efficacy, hopelessness and negative beliefs in depression among Lebanese individuals and thus to tackle the role of intolerance of uncertainty and of FoH in regard to this emotion.

**Open peer review.** To view the open peer review materials for this article, please visit http://doi.org/10.1017/gmh.2026.10178.

**Data availability statement.** All data generated or analyzed during this study are not publicly available due to the restrictions from the ethics committee but are available upon a reasonable request from the corresponding author (SH).

**Acknowledgements.** The authors would like to thank all participants.

**Author contribution.** F.F.-R., S.O. and S.H. designed the study; L.G. drafted the manuscript; S.H. carried out the analysis and interpreted the results; F.S. and M.D. collected the data. D.M. and A.M.A. reviewed the paper for intellectual content. All authors reviewed the final manuscript and gave their consent.

**Competing interests.** The authors have nothing to disclose.

**Ethics statement.** Ethics approval for this study was obtained from the ethics committee of the School of Pharmacy at the Lebanese International University (2024ERC-024-LIUSOP). Written informed consent was obtained from all subjects; the online submission of the soft copy was considered equivalent to receiving a written informed consent. All methods were performed in accordance with the relevant guidelines and regulations (in accordance to the declaration of Helsinki).

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
