## [Reviewer Report]

The study examines the potential mediating effect of fear of happiness (FH) on the relationship between intolerance of uncertainty (IU) and anxiety. While the direct impact is sufficiently described in the current literature, the possible role of FH remains unknown. Unfortunately, this argument of novelty is used throughout the article as the primary justification for this study, but I am concerned that it is insufficient. The originality of the research question and not exploring the described associations in Lebanon are not sufficient arguments to lead to the study. While they can be adequate for the investigation of relationships between all variables, the proposed mediation effect assumes a causal effect, which I see as very problematic. It can hardly be explained solely by the nature of the variables that the link should go from IO to FH and then to A. Firstly, IO can be considered as the most stable included variable, so I can agree that it can be at the beginning. On the contrary, from my point of view, both causal mechanisms are possible. While authors use the link going from FH to A, I think that the opposite (from A to FH) is possible. The introduction focuses mainly on associations and does not provide a clear argument for the proposed direction. Consequently, I would encourage authors to focus more on the chosen directions, explaining why they think this way and what the arguments are in support of it.

It is worth noting that the best option for testing the given mediation hypothesis would be a longitudinal study, which allows for the observation of the predicted causal effect. In the current design, only theoretical verification can be provided, not empirical. As it is probably not realistic, I am writing this mainly as a recommendation for future studies, as it does not seem (according to the Limitations section) that the authors are aware of it.

Besides this central point, there are several minor ones. For example, authors consistently use the word “prove”, but we must respect the limitations of psychological research in general (and this study in particular), and it would be better to use other, less strong words (e.g., indicate).

Next, the authors write: “The aim of the study was to collect data from 905 participants in Lebanon...” It is undoubtedly not the aim of this study, it is about methodology of answering research question.

"The Student’s t-test was used to compare a continuous variable and a dichotomous variable..." - it is not right, the t-test is used to compare the mean value of continuous. var. across two levels of dich. variable, not to compare these two variables.

The format of tables is not in line with the latest APA standards.

Correlations of variables with themselves = 1 are usually not displayed (use blank space or internal consistency coefficient).

The sentence in limitation interpreting the link between IO and FH as the result of lower happiness seems problematic - the mean value of one variable can hardly be used as justification for its relationship with other variables.

In summary, although I view the article as having serious shortcomings, I would recommend a major revision, which would provide an opportunity to improve it. Besides some formal and linguistic aspects, I recommend focusing mainly on the justification of the research question and, primarily, the mediation model, where the proposed causal mechanism is the weakest issue.

---

## [Reviewer Report]

Thanks for the opportunity to review the manuscript. The study is highly relevant, timely, and addresses a significant gap in global mental health research, especially concerning populations exposed to chronic crises. The methodology is generally sound, and the mediation analysis provides theoretical and clinical insights. Yet, there are minor revisions needed:

Abstract

1. For maximum clarity, consider mentioning the full name of the IU subscales (Prospective Anxiety and Inhibitory Anxiety) alongside the variable Intolerance of Uncertainty. The current description in the abstract uses the subscale names but refers to the overall construct (IU) in the title and conclusion

Introduction

2. A more explicit, brief theoretical justification for why FoH acts as a mediator (e.g., IU leads to hopelessness/negativity, which then manifests as FoH, causing greater anxiety by reducing self-efficacy) would strengthen the model introduction. (This detail is currently well-covered in the Discussion, but a preview in the Introduction would enhance the hypothesis foundation.)

Methods

3. Generalized Anxiety and FOH scales are not properly explained. For instance, the IUS-12 scale was described in order such of the number of items, Likert-scale, dimensions, and Cronbach’s alpha information for the scale. The latter scales were described very shortly. For instance, no information was provided on how the score is calculated and etc. Please be consistent throughout the measures and provide similar information across all measures and in a similar order.

Overall

4. Although the text is written well, there are typos, extra letters, and missing letters throughout the text, and those definitely distract from reading- the authors should revise the whole text.

---

## [Reviewer Report]

I appreciate the changes authors conducted. Unfortunately, one of my comments has not been understood. When I wrote that mediation requires (or at least assumes) causal effect, excluding words related to causality is not enough. My comment was more about the directionality of the path - if authors can provide a stronger justification for the link they proposed from FoH to anxiety, and that it does not go the opposite way. On the other hand, the added parts concerning various associations of FoH and A provide quite a clear picture supporting the proposed link, so I think that the whole mediation model is justified to the sufficient degree.

The misunderstanding resulted in some changes in discussion I think were not needed. My comments was not about the presence of causality, it can be expected in the mediation model, but about the justification of the proposed link (position of FoH and A). I am sorry for it, but as the Introduction section has been changed according to this, I think that the Discussion concerning the found effects could remain. Therefore, I would recommend minor revision, where the Discussion will be again more about causality (respecting the nature of cross-sectional study, as mentioned in limitations).

All other sections are OK.